# Viruses and Cajal Bodies: A Critical Cellular Target in Virus Infection?

**DOI:** 10.3390/v15122311

**Published:** 2023-11-25

**Authors:** Lucy Lettin, Bilgi Erbay, G. Eric Blair

**Affiliations:** 1School of Molecular and Cellular Biology, University of Leeds, Leeds LS2 9JT, UKerbay_bilgi@hotmail.com (B.E.); 2Moleküler Biyoloji ve Genetik Bölümü, Fen Fakültesi, Van Yuzuncu Yil University, Van 65140, Türkiye

**Keywords:** nuclear bodies, Cajal bodies, PML bodies, nucleolus, snRNPs, animal viruses, plant viruses, nuclear remodelling

## Abstract

Nuclear bodies (NBs) are dynamic structures present in eukaryotic cell nuclei. They are not bounded by membranes and are often considered biomolecular condensates, defined structurally and functionally by the localisation of core components. Nuclear architecture can be reorganised during normal cellular processes such as the cell cycle as well as in response to cellular stress. Many plant and animal viruses target their proteins to NBs, in some cases triggering their structural disruption and redistribution. Although not all such interactions have been well characterised, subversion of NBs and their functions may form a key part of the life cycle of eukaryotic viruses that require the nucleus for their replication. This review will focus on Cajal bodies (CBs) and the viruses that target them. Since CBs are dynamic structures, other NBs (principally nucleoli and promyelocytic leukaemia, PML and bodies), whose components interact with CBs, will also be considered. As well as providing important insights into key virus–host cell interactions, studies on Cajal and associated NBs may identify novel cellular targets for development of antiviral compounds.

## 1. Introduction

The eukaryotic cell nucleus is a highly organised yet dynamic intracellular environment in which most cellular DNA is condensed into chromatin and assembled into chromosomes. In addition, there are many structures that reside within the nucleus, known as nuclear bodies (NBs; Figure 1). NBs perform multiple functions, several of which are associated with gene expression and DNA replication. NBs are not bounded by membranes but are defined, both functionally and structurally, by the localisation and enrichment of nuclear factors. They have been referred to as biomolecular condensates [1]. For example, the Cajal body (CB) is defined by the presence of several key components, including p80-coilin [2,3,4], survival of motor neuron 1 (SMN [5]), TOE1 (target of EGR1) [6] and WRAP53 (WD40-encoding RNA antisense to p53) [7]. Functional links between CBs and other NBs exist. For example, fibrillarin and Nopp140 are present in both CBs and nucleoli, and Nopp140 mediates the shuttling of proteins between the two bodies [8,9,10,11].

CBs are targeted during infection by a number of plant and animal viruses, often in conjunction with the targeting of nucleoli and other NBs such as PML (promyelocytic leukaemia) bodies and nuclear speckles. Viruses frequently sequester, reorganise or degrade host cell components to facilitate their replication and establish systemic infections [12]. Furthermore, viruses may target NBs to inactivate antivirus functions that they may possess. This would be beneficial for the virus, for example, in the regulation of apoptosis by PML bodies [13]. Therefore, by studying the structure and function of NBs during virus infections, it may be possible to gain insights into the functions of NBs in uninfected cells. A deeper understanding of the interaction between viruses and NBs may open new therapeutic approaches to infections by targeting key cellular components or structures that facilitate virus replication.

In this review, current research on NBs will be considered, with a particular focus on CBs and the viruses that target them. There is an often-recurring motif of nuclear architecture disruption in virus-infected cells, particularly in CBs, which provides evidence for functional associations between NBs and infection in virus-infected eukaryotic cells.

### 1.1. Cajal Bodies

Cajal Bodies (CBs), also termed coiled bodies, were first discovered and characterised as nucleolar accessory bodies (named due to their localisation with the nucleolar periphery) in vertebrate neuronal cells by Ramon y Cajal in 1903 [14]. CBs are localised in the nucleoplasm and are approx. 1 µm in diameter. Under the light or electron microscope, CBs appear as a structured tangle of fibrillar threads [15]. There are generally between 1 and 6 CBs per cell, dependent on the cell type [16]. CBs are ubiquitous in foetal tissues. In these tissues they display greater distinction from the closely-associated NB, a Gemini of coiled bodies (gems), whereas in later development or in adult tissues, the two bodies are more closely associated and, in some cases, indistinguishable [17]. However, both CBs and gems are less common in adult tissues. Their abundance seems to decrease with differentiation, and during later development they are more apparent in metabolically active cells [18]. CBs have been detected in motor neuron cells, where they have a larger volume. However, they are present in all cells in the central nervous system except blood vessels. They have been detected in epithelial cells of the liver, pancreas, duodenum and oesophagus, skeletal muscle, Islets of Langerhans, as well as liver parenchymal cells [18].

CBs are mobile bodies and traverse the nucleoplasm of both plant and animal cell nuclei, including to and from the nucleolus. They are capable of both fusing together and dividing into smaller bodies [19,20]. This discovery infers the existence of a regulatory or feedback system that functions to determine the volume and number of CBs in each nucleus. The volume and number of CBs per cell is also dependent on the stage of the cell cycle. For example, CBs are most abundant in the late G1 phase but appear smaller in volume. In the S and G2 phases, the numerous smaller CBs that exist in G1 appear to fuse into fewer, larger CBs. During mitosis (M phase), no CBs are observed, although levels of cellular p80-coilin remain constant [21]. This suggests that CBs disassemble during the M phase and reassemble at the beginning of the interphase.

The assembly of CBs is dependent on VRK1, an abundant nuclear serine-threonine protein kinase. VRK1 phosphorylates p80-coilin at two serine residues, Ser184 and Ser489. VRK1-mediated phosphorylation protects p80-coilin from ubiquitination and subsequent proteasomal degradation following CB disassembly during mitosis [22].

CBs respond to cellular stress and can be disrupted by UV-C irradiation, causing some of their components, e.g., p80-coilin and snRNPs, to be displaced and redistributed in the nucleus. However, some factors, including the fibrillarin and Nopp140 proteins, remain in CBs under these conditions [23]. CBs are also linked to ageing through their colocalisation with telomerase RNA, which is a key component of the telomerase complex (TERC), particularly during the S phase [16]. TERC is a ribonucleoprotein enzyme responsible for stabilising chromosomes by adding DNA repeat sequences (telomere repeats) to their ends. Shortening of telomeres over cell generations due to reduction in TERC activity is critical for the process of cell senescence in normal cells, a process that is subverted in cancer cells [24].

Plant CBs contain several homologues of animal CB components [25], such as p80-coilin [2], small nuclear RNP (snRNP) maturation and splicing factors such as SMN [26] and small nucleolar RNP (snoRNP) maturation factors such as fibrillarin [27]. Furthermore, plant CBs facilitate different aspects of RNA processing, including assembly and modification of the spliceosome [28,29], as well as containing small Cajal body-specific RNAs (scaRNAs) and small nucleolar RNAs (snoRNAs) that function in RNA metabolism, affecting plant growth and development [30]. In *Arabidopsis thaliana*, components of the plant gene silencing machinery, such as Argonaute 4, colocalise with CBs, suggesting a role for CBs in the siRNA transcriptional silencing pathway [31]. The dynamics and mobility of CBs also exist in plants. In *A. thaliana* and tobacco BY-2 cells, CBs fuse, fragment and traverse the nucleoplasm in a similar fashion to animal cells [19]. Plant CBs also perform unique functions that are not present in animal cells, presumably due to the necessity to respond to physiological stresses and conditions unique to plants [32].

### 1.2. The Nucleolus

The nucleolus comprises three compartments, the fibrillar centre (FC), the dense fibrillar component (DFC) and the granular component (GC) [33]. Genes that encode ribosomal RNAs (rDNA) and their rRNA transcripts, as well as many nucleolar proteins, are localised within nucleoli [34]. The three compartments and associated proteins correspond to the major functions of the nucleolus. For example, pre-rRNA is transcribed from rDNA in the FC or at the border between the FC and DFC [35]. Thus, the RNA polymerase I machinery, including DNA topoisomerase and RNA polymerase I, is localised in the FC [36,37]. Pre-rRNA processing factors associate with the DFC, including fibrillarin [38], and the proteins of snoRNPs. The GC envelopes the FC and DFC and it is here where the predominant function of the nucleolus occurs, as the site of pre-ribosomal subunit biogenesis [35]. Chromatin localises with the nucleolus or perinucleolar regions; however, this colocalisation of chromatin may reflect a functional role for the nucleolus in silencing gene transcription at particular loci [39].

The nucleolus has also been linked with ageing [40], cell cycle control [41,42], preventing initiation of DNA replication following the cellular stress response and the generation of RNPs [43,44,45]. A regulatory role for the nucleolus in DNA damage repair has been proposed, as well as in other cellular and nuclear processes [46]. During cellular stress caused by DNA damage, the nucleolus undergoes segregation of its nucleolar components. This was first observed in cells in which topoisomerase II was chemically inhibited, resulting in a reduction in rRNA synthesis [47]. Nucleolar segregation (the process of movement of nucleolar proteins into the nucleoplasm following cellular stress) was subsequently observed in cells in which DNA had been damaged by UV irradiation [48]. The nucleolus was further linked to apoptosis in UV-irradiated cells [49]. Rearrangement of nucleolar architecture has also been associated with cell cycle checkpoints. For example, prior to mitosis, the nucleolus is disassembled and then reassembled at the end of mitosis by the relocalisation of its core components [50].

### 1.3. Promyelocytic Leukaemia (PML) Bodies

Promyelocytic leukaemia (PML) bodies are composed of an outer insoluble scaffold surrounding an inner core of over 50 constitutively and transiently associated proteins, including the protein product of the p53 tumour suppressor gene. The PML protein accumulates in PML bodies and is often used as a marker protein for this NB [51]. PML bodies cannot assemble in the absence of SUMO-1-modified PML protein, which is essential for the colocalisation of several proteins with PML bodies [52]. PML belongs to a family of proteins that are defined by the presence of an RBCC (RING-B-box-coiled-coil) motif [53]. This motif consists of a RING finger (C3HC4 zinc finger), which forms tetramer torus structures, B-boxes composed of one or two cysteine rich regions and a coiled-coil formed from a leucine chain [13,52].

PML bodies respond to cellular stress by undergoing changes in their native architecture and, in doing so, regulate stress-induced sumoylation, the process whereby a small ubiquitin-like modifier (SUMO) protein is covalently attached to target proteins, altering their biological functions [54]. Ablation of PML in mice has shown that PML is a regulator of haemopoietic differentiation, growth and tumorigenesis. This discovery led to the PML protein being recognised as a tumour suppressor, specifically in acute promyelocytic leukaemia, due to the close association of the PML gene with the retinoic acid receptor-alpha (RAR-α) gene in this disease type [13]. PML is also critical for multiple apoptotic pathways including Fas and caspase-dependent DNA-damage induced apoptosis [55]. PML bodies also have a functional role in antimicrobial defence, since biosynthesis of the PML protein is induced by all the major interferon (IFN) types (α, β and γ), and overexpression of PML inhibits the replication of several viruses, including influenza and vesicular stomatitis viruses [56].

### 1.4. Components of Cajal Bodies

#### 1.4.1. p80-coilin

CBs are characterised by the enrichment of a marker protein p80-coilin (a nuclear autoantigen of molecular mass 80 kDa in humans) [57,58]. Although p80-coilin is present in the nucleoplasm, antibody staining for p80-coilin and subsequent immuno-electron microscopy showed that the protein localises to distinct bodies, corresponding to CBs [57]. Germline knockout of p80-coilin in mice [4], zebrafish [3] and *A. thaliana* [2] revealed that p80-coilin has an important role in the assembly and integrity of CBs, as well as for growth and development of the organism. For example, during gestation in p80-coilin knockout mice, there was prenatal semi-lethality in homozygous animals, which suggested that, while p80-coilin is important for development, it is not an essential protein for the mouse [59]. In addition, as previously mentioned, p80-coilin is phosphorylated at two amino acids during mitosis when CBs are disassembled, which further indicates that p80-coilin is associated with the structural integrity of CBs [60].

p80-coilin is a self-interacting protein, with serine phosphorylation sites and two nuclear localisation signals (NLS) located in the central domain. However, these consensus targeting motifs are not sufficient to target p80-coilin to CBs but are responsible for targeting p80-coilin to the nucleus [61]. The self-interacting domain of p80-coilin has been mapped to within the N-terminal 92 amino acids. p80-coilin self-interaction activity is reduced by hyperphosphorylation of the p80-coilin protein [62]. Furthermore, p80-coilin has a cryptic nucleolar localisation signal (NoLS) (Figure 2), which may be indicative of the protein’s involvement in a specific, as yet undefined, cellular response, which is also regulated by phosphorylation [62]. The C-terminus of p80-coilin is conserved and folded into a Tudor domain. Tudor domains have been shown to have preferential binding to DNA, RNA and modified amino acids. p80-coilin also contains an RG-box, a region enriched in arginine and glycine (Figure 2). The RG-box has been shown to form direct interactions with the survival of motor neuron (SMN) protein, thus localising SMN to CBs [63].

#### 1.4.2. SMN

The SMN protein is encoded by the survival of motor neuron 1 gene (SMN1) and, despite its name, is found in all cells [64]. Gems (gemini of coiled bodies), which often colocalise with CBs, particularly in non-foetal tissues, are characterised by the enrichment of SMN protein [5]. SMN is also localised in CBs in many cell types [18]. The SMN protein is critical for the formation of the SMN complex which, along with seven Gemin proteins, is required for spliceosomal snRNP maturation [65]. This might explain the functional and spatial overlapping of snRNP-enriched Cajal bodies with gems. SMN and the SMN complex function in the pre-mRNA splicing cycle and splicing complex recycling [66]. Overexpression of SMN leads to the disruption of CBs, suggesting a potential regulatory function for the protein in defining the number and volume of CBs within the nucleus [67].

Reduced abundance of SMN protein results in the neuromuscular disorder spinal muscular atrophy (SMA) [68]. The disorder is caused by point mutations or deletions in the SMN1 gene. Symptomatic phenotypes in individuals suffering with SMA can be severe. This includes loss of motor neurons, muscle weakness, immobility, respiratory failure and death. It is thought that SMA results from a disturbance in snRNP assembly caused by the loss of SMN’s chaperoning capabilities [68]. Furthermore, it has been demonstrated that SMN directly interacts with p53, both of which colocalise with CBs, suggesting the potential for an additional role for CBs in the regulation of cellular stress [69].

#### 1.4.3. TOE1

Target of *Egr1* (TOE1) is a highly conserved protein with roles in cell proliferation (maintenance of cellular p21, an inhibitor of cell proliferation) [6] and is a target of early growth response 1 (EGR1) protein [70]. It colocalises with p80-coilin and SMN in CBs. TOE1 is important for the integrity of CBs, acting in concert with p80-coilin and TOE1 knockdown, resulting in reduced SMN recruitment to CBs, with the protein instead accumulating in cytoplasmic foci. This suggests that TOE1 may be involved in the localisation of SMN to CBs. TOE1-deficient cells also showed reduced cell proliferation and splicing capabilities [6].

#### 1.4.4. WRAP53

The WD40 encoding RNA antisense to p53 (WRAP53) gene encodes a natural antisense transcript of p53 (WRAP53a) that regulates endogenous p53 mRNA [71], as well as a scaffold WD40 protein, WRAP53P (also known as WRAP53). This latter protein functions to guide factors to CBs as well as to telomeres and DNA double-strand breaks [71,72,73]. WRAP53 is critical for CB maintenance as well as for directing SMN to CBs. Knockdown of WRAP53 was disruptive to existing CBs in the nucleus, preventing assembly of new CBs, and resulting in the relocation of SMN and p80-coilin to the nucleolus. This highlights the importance of WRAP53 to the integrity of CBs [7]. Overexpression of WRAP53 is disruptive to CBs, resulting in their disassembly and the displacement of p80-coilin and SMN to the nucleoplasm [7].

#### 1.4.5. Nopp140 and Fibrillarin

Nopp140 (nucleolar phosphoprotein 140) and fibrillarin are protein components of CBs. p80-coilin directly interacts with Nopp140, which shuttles between the nucleolus and CBs, as well as between the nucleolus and the cytoplasm, inferring a dynamic relationship between the nucleolus and CBs [8,10]. Nopp140, along with fibrillarin, is also found in the nucleolus, which suggests a functional relationship between the two nuclear bodies. Fibrillarin, a highly conserved nucleolar protein, interacts with the SMN protein [9,11]. Furthermore, both Nopp140 and fibrillarin interact with snoRNPs, with fibrillarin-binding C/D box snoRNAs and Nopp140 binding C/D and H/ACA snoRNAs [16].

#### 1.4.6. snRNPs, snoRNPs and scaRNPs

Almost all small non-coding RNAs (sncRNAs) present in the cell, except tRNAs and miRNAs, directly interact with p80-coilin [74]. It is widely accepted that CBs are critical in the biogenesis of snRNPs; however, beyond this function, the association with snRNAs implies that CBs may also act as a central hub or coordinator for several processes in RNA metabolism.

Spliceosomal snRNPs are closely associated with CBs in both plant and animal cells. They are composed of a small nuclear RNA (of approx. 150 nucleotides), 1–12 specific proteins per snRNP and an Sm protein, or Sm-like protein, in a heptameric ring. Currently, five major snRNPs (U1, U2, U4, U5 and U6) and four minor snRNPs (U11, U12, U4atac and U6atac) have been identified based on the presence of the snRNA [75]. The intracellular distribution of snRNPs, detected using fluorescently labelled antisense RNA probes, revealed colocalisation of snRNPs and snRNAs with specific nuclear foci that were later defined as CBs. Certain snRNPs also colocalise with the nucleolus. Furthermore, U1 snRNA is distributed throughout the nucleoplasm and, although it colocalises with CBs, it is not concentrated there [76,77]. The two major modifications to snRNAs are pseudouridylation and 2′-O-methylation. These modifications are guided by two classes of snoRNPs, which are defined by their snoRNA component. Box H/ACA snoRNPs contain dyskerin that completes pseudouridylation and C/D snoRNPs contain fibrillarin, which effects ribose methylation [78].

Small CB-specific RNAs (scaRNAs) form a family of eight snRNAs that colocalise specifically with CBs. They direct the pseudouridylation and ribose methylation modifications of U1, U2, U4 and U5 snRNAs, which are transcribed by RNA polymerase II [79,80,81]. In contrast, U6 RNA, transcribed by RNA polymerase III, undergoes the same modifications in the nucleolus by alternative snRNAs [82,83]. The H/ACA scaRNAs (and C/D-H/ACA mixed domain scaRNAs) are targeted to CBs by the *cis*-acting sequence, the CAB box, of which H/ACA has two copies [84]. Furthermore, a conserved WD40 protein, originally isolated in *Drosophila melanogaster*, termed WDR79 in humans (also known as WRAP53: see Section 1.4.4), is required for the localisation of scaRNAs in CBs. It functions by binding to cellular scaRNAs, acting in a CAB box-dependent interaction [85].

snRNP biogenesis is a multi-step process that takes place in the cytoplasm and nucleus of animal and plant cells. The complete pathway has been reviewed [86]. snRNPs colocalise in CBs during their generation cycle, prior to translation, processing and assembly of the SMN complex in the cytoplasm. Furthermore, the integrator complex, which is involved in the initial cleavage to form pre-snRNA and terminal 3′-end processing of snRNAs [87], is critical for CB integrity and homeostasis [88]. Following assembly, the import complex is transported into the nucleus where it again passes through CBs [86,89]. This localisation to CBs following import coincides with modifications of snRNPs, including 2′-O-methylation and pseudouridylation, implying a functional role for CBs in the biogenesis pathway, a process guided by scaRNAs [90]. By binding its associated proteins, CBs facilitate snRNP assembly by bringing the factors in close enough proximity to interact. CBs are involved in the generation of U4/U6 di-snRNPs and U4/U6.U5 tri-snRNPs [86]. The integrator complex mediates processing of snRNAs. An activity-deficient integrator complex had a negative impact on the colocalisation of CB core components. This included a depletion of snRNPs in CBs, the relocalisation of some p80-coilin to the nucleoplasm (subsequently forming CB-like foci) or nucleoli and SMN to the cytoplasm [88].

snRNP biogenesis is linked to growth and development as well as stress responses in plants. An example of this is the diverse developmental-defective phenotypes observed in snRNP-mutant *A. thaliana*, including effects on female gametogenesis and flowering, as well as enhanced stress sensitivity [91].

## 2. Viruses That Interact with Nuclear Bodies

There is much yet to be discovered concerning the function and mechanisms of NBs and how they contribute to the complex system of nuclear and cellular functions. In both animals and plants, certain viruses target these bodies, pointing towards potentially conserved cellular targets of diverse viruses. Investigation into the mechanism(s) of action and functional consequences of virus targeting of NBs may provide further insights into our understanding of the cell biology of NBs, virus–host cell interactions and the pathogenesis of infections.

### 2.1. Animal Viruses Interacting with Nuclear Bodies

This section will discuss how viruses that infect members of the animal kingdom interact with NBs, in particular with Cajal bodies. Viruses that interact with Cajal bodies and other NBs are summarised in Table 1.

### 2.2. Animal Viruses That Target Cajal Bodies

#### 2.2.1. Minute Virus of Mice

Minute virus of mice (MVM) is an autonomous parvovirus that possesses a linear ssDNA genome. The genome encodes three major classes of mRNA, R1-3. The virus capsid proteins VP-1 and VP-2/3 are encoded by R3 while the two non-structural polypeptides, NS1 and NS2, are encoded by R1 [114,115]. MVM, in a similar manner to the related parvovirus H-1 [116], has the capability to form novel subnuclear structures, termed autonomous parvovirus-associated replication (APAR) bodies, during infection, which are distinct from host nuclear bodies. They appear to be sites of virus replication due to the colocalisation and accumulation of replication factors [117]. NS1 specifically interacts with host SMN protein and colocalises with CBs [100]. This interaction was detected late in infection, coinciding with major virus-induced nuclear reorganisation, including components from CBs and APAR bodies, which reconstitute into multiple large nuclear bodies termed SMN-associated APAR bodies (SAABs). NS1 is critical for virus replication and transcription regulation and its accumulation in SAABs suggests that these bodies may be important in the virus life cycle and/or may indicate the initiation of cell death mediated by the virus via NS1 cytotoxicity [100].

#### 2.2.2. African Swine Fever Virus

African swine fever virus (ASFV) is the only member of the *Asfaviridae* genus within the nucleocytoplasmic large DNA viruses superfamily and causes severe disease in pigs [108]. It possesses a dsDNA genome containing between 150 and 167 ORFs [109]. As well as other pathogenic effects within the host nucleus, ASFV infection causes the structural rearrangement of host NBs. In terms of nuclear speckles, ASFV infection results in their enlargement and reduction in number per nucleus [108]. Depletion of PML resulted in malformed virus factories and a reduction in progeny virus, indicating that PML facilitates ASFV infection. In addition, PML bodies were disrupted during virus infection, increasing in size and decreasing in number [108]. CBs are remodelled during infection by ASFV, being fragmented into a greater number of ‘comma-shaped’ bodies [108]. This virus-induced structural reorganisation of host nuclear bodies may contribute to ASFV control of host transcriptional machinery [108,110].

### 2.3. Human Viruses Interacting with Cajal Body Components to Induce Antivirus Defence

#### Human Immunodeficiency Virus

Human immunodeficiency virus (HIV) isolates are classified into two types, HIV-1 and HIV-2, which differ in the organisation of their genome, with HIV-1 being the most prevalent worldwide. It is a member of the Lentivirus genus within the *Retroviridae* family. The HIV genome comprises two identical positive-sense ssRNA molecules [112]. TOE1, a conserved multifunctional nuclear protein that is localised to CBs, inhibits replication of HIV-1 at the transcriptional level. This occurs by direct interaction between TOE1 and the HIV-1 TAR RNA sequence element, thus interfering with Tat transactivation, which is critical for the virus RNA elongation step of transcription. This interaction occurs via a lysine/arginine-rich NLS domain in TOE1. It has been proposed that TOE1 acts to competitively bind TAR in place of Tat to inhibit RNA transcription of the virus genome [111]. TOE1 has also been linked to the host CD8^+^ T cell-mediated immune response [111]. HIV-1 also targets other NBs. For example, HIV-1 RNA and complementary DNA are located in nuclear speckles in infected human macrophages while, in infected human T cells, a cellular long non-coding RNA (lncRNA) NEAT1 regulates HIV-1 production and is located in paraspeckles [113].

### 2.4. Human Viruses Interacting with Cajal Body Components

#### 2.4.1. Influenza A Virus

Influenza A virus (a member of the *Orthomyxoviridae* family) is one type of influenza virus (the other less common types being B and C) that cause recurrent epidemics and occasional pandemics in humans and other mammalian and avian species. Their persistence in these populations is largely due to continuous evolution of the viruses, which is driven in influenza A by the natural virus gene reservoir in aquatic birds [118]. As in other influenza viruses, influenza A has a negative-sense ssRNA genome, which is segmented into eight RNAs that exist as virus ribonucleoproteins (vRNPs) in the virus capsid [104,118,119]. Replication and transcription of the virus genome takes places in the host nucleus, by sequestering components of the nuclear machinery [120]. Influenza A targets NBs in several ways, including the accumulation of non-structural protein 1 (NS1) and matrix protein (M1) in PML bodies in order to interfere with the host interferon-induced antivirus pathway, thus suppressing apoptosis [107]. Overexpression of the PML protein resulted in cellular resistance to influenza virus infection [121]. The host nucleolus is remodelled during influenza A infection, in association with virus replication [122,123]. This restructuring of the nucleolus was found to be mediated by the NS1 and M1 virus proteins, and may subvert host nucleolar functions [123]. Influenza A nucleoprotein (NP) and NS1, the latter of which interacts with nucleolin, localise to the nucleolus at different times during infection via nucleolar localisation signals (NoLs) present within the virus proteins [102,105,106]. As well as colocalisation with nucleoli, NP has been shown to tightly associate with regions of the nucleus adjacent to PML bodies and CBs [104]. Influenza A virus infection causes an increase in the number of CBs per nucleus and a concurrent reduction in their size [103]. Influenza A NP colocalises with CBs, tightly associating with them, and in doing so follows the same rearrangement as CBs. It was proposed that this association between NP and nucleoli and CBs, as well as the relative fragmentation of both, may be related to trafficking between the two NBs or there may be a further functional link [104].

#### 2.4.2. Zika Virus

Zika virus (ZIKV) is an arthropod-borne virus (arbovirus) within the *Flaviviridae* family (Flavivirus genus) of viruses and has a positive-sense ssRNA genome [99]. The virus RNA genome encodes a single polypeptide, which is subsequently proteolytically cleaved to produce three structural proteins (capsid (C), envelope (E) and membrane precursor (prM)) and seven non-structural proteins (NS1, NS2A, NS2B, NS3, NS4A, NS4B and NS5) [124]. ZIKV proteins target cytoplasmic bodies, including the endoplasmic reticulum and lysosomes, as well as nuclear bodies [99]. The ZIKV NS5 protein associates with host CBs via interaction with p80-coilin, causing disruption of CBs [99]. Furthermore, NS5 expression coincided with an increase in the number of CBs per nucleus and a corresponding reduction in their volume [99].

#### 2.4.3. Adenoviruses

There are at least 100 types of human adenoviruses (Ads), classified into seven species, A-G [125]. The adenovirus genome is a linear double stranded DNA molecule. It encodes five early transcriptional units (E1A, E1B, E2, E3 and E4), four intermediate transcription units (IX, IVa2, L4 intermediate and E2 late) and one major late transcriptional unit (MLTU). Five families of late mRNAs (L1-5) are generated from a single pre-mRNA by differential pre-mRNA splicing. A common tripartite leader sequence is spliced to the coding regions for mRNAs encoding structural proteins of the virus and non-structural proteins that have functions in virus assembly and transcriptional regulation [125].

Ad5 infection results in the redistribution of CBs into smaller foci termed rosettes or microfoci, as defined by immunofluorescence with anti-p80-coilin [95,97,98]. The microfoci appear to be unique structures. They occur adjacent to, but do not colocalise with, virus DNA replication centres (containing the virus E2-72K protein), areas of virus pre-mRNA splicing and mRNA export (containing the ASF/SF2 protein) [97] or with Aly, a component of the major mRNA export TREX complex [98]. The formation of these CB microfoci takes place at the onset of Ad5 DNA replication and at the intermediate/late phase of infection; the blocking of Ad DNA replication prevents CB reorganisation [97]. CBs are associated with a population of nuclear actin in uninfected cells and in the early phase of Ad5 infection. However, late in Ad5 infection, actin disassociates from CBs as they are rearranged, suggesting that actin is involved in the formation of CB microfoci [96]. p80-coilin facilitates Ad5 infection as, following p80-coilin siRNA-mediated reduction, virus yield is reduced by up to 80% and the levels of most virus proteins are reduced in p80-coilin siRNA-treated cells compared to controls [98]. The underlying mechanism in the reduction in virus protein and virus production in p80-coilin-depleted cells appears to be a decrease in export of Ad5 mRNA transcripts from the nucleus to the cytoplasm [98].

Immunofluorescent antibody staining of Ad5-infected cells revealed colocalisation of p80-coilin in microfoci, with areas of immunoreactivity defined by an antibody that recognised the virus L4-22K and L-33K proteins. The L4-22K protein shares an N-terminal domain with L4-33K but has a unique carboxy-terminal region due to differential pre-mRNA splicing. The L4-22K protein has multiple functions in Ad infection, including transcriptional and post-transcriptional regulation of virus gene expression [126,127]. Co-immunoprecipitation analysis showed that only the L4-22K protein formed a stable complex with p80-coilin in Ad5-infected cells and in cells co-transfected with plasmids encoding epitope-tagged p80-coilin and either the L4-22K or -33K protein [98].

Overall, the data described above suggest that, during the late phase of Ad5 infection, CBs are reorganised into microfoci by a currently unknown mechanism involving nuclear actin. One consequence of production of microfoci may be an increase in the surface area of the CB and/or the release of CB proteins such as p80-coilin into the nucleoplasm. The virus L4-22K protein forms a complex with p80-coilin, potentially facilitating and directing virus transcripts to complexes such as Aly/Ref/TREX or CRM1 for mRNA export [98].

PML bodies are reorganised in Ad-infected cells [94]. The two major E1A proteins colocalise with PML bodies in Ad-infected cells [94]. Furthermore, the E4-ORF3-encoded 11 kDa protein appears to be the sole protein responsible for the reorganisation of PML bodies during infection [94]. E4-ORF3-dependent rearrangement of PML bodies has been shown to activate PML-dependent, interferon-induced antivirus innate immune defence [128]. Infection of cells by Ad5 also induces the redistribution of the nucleolus, involving the displacement of chromatin and several nucleolar components, including fibrillarin [95].

#### 2.4.4. Herpesviruses

The herpesviruses form a large group of DNA viruses causing a range of infections in humans and animals. They are divided into three sub-families, alpha (e.g., herpes simplex virus 1, HSV-1), beta (e.g., Cytomegalovirus, CMV) and gamma (Epstein–Barr virus, EBV), all of which have a large coding capacity of between 80 and 200 proteins. They can establish latent and lytic infections and have different cell tropisms and life cycles. In a large-scale, genome-wide screen of proteins encoded by these viruses, several expressed ORFs were found to associate with or disrupt CBs, PML bodies or nucleoli [101]. Ninety-three virus proteins out of 269 recovered as molecular clones encoded by the three viruses were located in the nucleus. No virus protein associated with CBs; however, cells expressing the CMV UL3 or UL30 ORFs contained significantly fewer CBs compared to control cells, suggesting that these virus gene products disrupt CBs. The avian alphaherpesvirus, MDV (Marek’s disease virus), encodes oncogenic proteins Meq and a fusion of Meq with a virus homologue of interleukin 8, Meq/vIL8. In cells transfected with either construct (expressed as a fluorescent fusion protein), around 20% of cells showed colocalisation of p80-coilin with either Meq or Meq/vIL8, with generalised nuclear or nucleolar localisation found in the remainder of the cells [92,93]. The CMV UL3, UL30, and the Meq gene products associated with or reorganised PML bodies, providing further evidence of the inter-relationships between these two NBs [92,93,101].

### 2.5. Plant Viruses Interacting with Nuclear Bodies

This section will discuss how viruses that infect members of the plant kingdom interact with NBs, in particular with Cajal bodies. Viruses that interact with Cajal bodies and other NBs are summarised in Table 2.

#### 2.5.1. Virus Interactions with the Cajal Body Component, Fibrillarin

##### Potato Virus A

Potato virus A (PVA) belongs to the Potyvirus genus (family *Potyviridae*). It is part of a large group of positive-sense ssRNA viruses like picornaviruses in animals. In order to generate proteins, of which they can produce up to ten, a large polyprotein is translated and cleaved by three virus-encoded proteinases [132]. The multifunctional virus protein nuclear inclusion protein-a (NIa) is targeted to CBs [132]. NIa is conserved among several other plant viruses including tobacco etch virus (TEV), also belonging to the *Potyviridae* family [146]. NIa is a polyprotein with a C-terminal proteinase domain and an N-terminal virus genome-linked protein (VPg) domain, the latter of which contains the NLS required for colocalisation of the protein with the host nucleus [146]. In PVA, the VPg domain of NIa was found to contain both nuclear and nucleolar localisation signals [132]. Furthermore, this same domain, more specifically NLS I (amino acids 4–9), was shown to target and direct the accumulation of NIa to CBs, where the VPg domain interacted with fibrillarin. The functional consequences of this interaction were demonstrated by the depletion of fibrillarin, which resulted in reduced infectivity of PVA in *Nicotiana benthamiana* (tobacco) plants [132].

##### Citrus Tristeza Virus

*Citrus Tristeza Virus (CTV)* is a member of the *Closteroviridae* family and contains a large, approx. 20 kb, ssRNA positive-stranded genome that encodes approx. 17 proteins. The CTVp23 protein is a major pathogenicity determinant of CTV, acting by RNA silencing suppression. CTVp23 accumulates in Cajal bodies and in nucleoli, where it colocalises with fibrillarin [129]. Expression of CTVp24 alone recapitulates many of the pathogenic properties of the whole virus and formed the basis for mutational analysis of the protein. This identified a bipartite nucleolar localisation sequence and revealed that most of the CTVp23 sequence is required for CB and nucleolar localisation [129].

##### Groundnut Rosette Virus

Groundnut rosette virus (GRV) is a member of the Umbravirus genus (*Tombusviridae* family), and has a genome consisting of a linear segment of positive-sense ssRNA. The genome is divided into four open reading frames (ORF1-ORF4). ORF1 and ORF2 are located at the 5′ end, slightly overlapping with each other, and are translated by a frameshift mechanism as a single fusion protein. Following ORF2 is a short untranslated region, which separates it from ORF3. Finally, ORF3 and ORF4 are located at the 3′ end and overlap almost entirely [147]. ORF4 encodes a 28 kDa protein involved in cell-to-cell movement and localises to, or in close proximity to, the plasmodesmata (channels connecting neighbouring plant cells), thus fulfilling a role similar to movement proteins (MPs) in other plant viruses [148]. ORF3 encodes a 27 kDa protein which is involved in long distance movement and stabilisation of virus RNA [147,149]. ORF3 may utilise existing trafficking pathways through the nucleolus and involving the rearrangement of CBs. Once the ORF3 protein enters the nucleus it targets CBs and reorganizes them into multiple CB-like structures (CBLs), potentially by interacting with an SMN homologue [133]. This reorganisation probably occurs by displacing CB components, such as coilin and fibrillarin, and redistributing them with ORF3 into the new CBL structures. ORF3 then localises to the nucleolus by fusing CBLs with the nucleolus in a novel pathway [133]. This fusion of CBLs with the nucleolus relies on the interaction of ORF3 with fibrillarin, shown by silencing the fibrillarin gene, which prevented long-distance, but not cell-to-cell, movement [134]. Furthermore, reduction in the expression of fibrillarin caused ORF3 to accumulate in CBLs, which were unable to fuse with the nucleolus [134]. However, the ORF3–fibrillarin interaction does not appear to be critical in either targeting of ORF3 to CBs or in their reorganisation [134]. There is a further function of ORF3, in which ORF3 redistributes fibrillarin to the cytoplasm in order to be used for RNP particle generation [134]. ORF3 interacts with virus RNA and subsequently forms filamentous RNP particles, protecting the virus RNA during long distance transport through the host phloem, thus aiding virus spreading in the plant [134,150]. Another member of the *Tombusviridae* family, Pelargonium line pattern virus (PLPV), encodes a p37 protein that interacts with p80-coilin and fibrillarin. The p37 protein is a capsid protein that may be involved in the suppression of RNA silencing, thus overcoming plant host defence [131].

#### 2.5.2. Virus Interaction with the Cajal Body Protein Coilin, Resulting in Increased Pathogenicity

##### Poa Semilatent Virus

Poa semilatent virus (PSLV) is a member of the Hordeivirus genus and has a positive-sense RNA genome [151]. The genome comprises three RNAs (α, β, and γ). RNAβ encodes the first ORF encoding the coat protein, followed by several overlapping ORFs, which together form the triple gene block (TGB). The two further ORFs encode small hydrophobic proteins [152]. The first gene of the triple gene block (TGBp1) encodes the movement protein (MP) of PSLV, which interacts with CBs during virus infection. TGBp1, analogously to GRV ORF3, is involved in interacting with virus RNAs to form RNP complexes for cell-to-cell movement, which is also critical for long distance transport and establishing a systemic infection in the plant [138,139]. A direct interaction between coilin and PSLV MP was shown both in vitro using purified recombinant proteins and in vivo, following introduction of a TGBp1 expression construct into *N. benthamania* plants. Deletion analysis revealed a major coilin binding site in the positively-charged N-terminal domain of TGBp1 [139].

#### 2.5.3. Virus Interaction with Cajal Body Marker Protein Coilin, Resulting in Decreased Pathogenicity

##### Tobacco Rattle Virus

The involvement of coilin in tobacco rattle virus (TRV) infection produces a different outcome than in most plant viruses in that it has a paradoxical role in host *N. benthamiana* antivirus defence. TRV is a bipartite (RNA1 and RNA2) virus, belonging to the Tobravirus genus (*Virgaviridae* family), with a positive-sense ssRNA genome [153]. RNA1 encodes a 29–30 kDa MP in the 1a ORF and a 12–16 kDa cysteine-rich silencing suppressor protein (termed 16K) in the 1b ORF. Both 1a and 1b are translated from subgenomic RNA (sgRNA) [153]. RNA1 also encodes the helicase and RNA-dependent RNA polymerase (RdRp) proteins. RNA2, the smaller of the two RNAs, encodes the virus coat protein as well as several proteins involved in virus transmission to nematodes [153]. TRV-infected wild-type (WT) plants are able to recover from infection [140], suggesting the existence of a host antivirus pathway. The host defence system comprises both RNA silencing and salicylic acid (SA) pathways, which require the interaction of the virus 16K protein and host coilin. This was demonstrated by the knockdown of host coilin or deletion of 16K in the virus, which prevents plant recovery and results in an exacerbation of symptoms [141]. RNA silencing is a major plant antivirus response. In this case, it was demonstrated in coilin knockdown plants where, contrary to expectation, depletion of coilin increased TRV accumulation and TRV-specific small interfering RNAs (siRNAs), suggesting the RNA silencing pathway is negatively affected by coilin knockdown [140]. However, the discovery that RNA silencing mechanisms can persist in coilin-deficient plants suggested that another host antivirus defence pathway was also involved [140]. In terms of the SA-dependent antivirus response against TRV, coilin knockdown and/or deletion of the virus 16K resulted in similar effects, namely the accumulation of SA and expression of SA-responsive genes. This suggests that the two mechanisms are working together to produce an antivirus response leading to plant recovery [140,141]. A stable interaction of 16K and coilin has been demonstrated in vitro, as well as the nucleolar localisation of coilin via interaction with 16K in WT plants [141]. A molecule, poly(ADP-ribose) polymerase 1 (PARP-1), that links the processes of SA-activated gene expression and plant resistance to TRV infection has recently been identified and can also participate in the coilin/16K complex [154].

##### Barley Stripe Mosaic Virus

Barley stripe mosaic virus (BSMV) induces plant antivirus responses in a similar way to TRV. BSMV is a member of the Hordeivirus genus (*Virgaviridae* family) and has a tripartite (RNAα, β and γ) positive-sense ssRNA genome [155] encoding seven proteins [152]. RNAβ contains four ORFs (βa-d) with βa encoding for the virus coat protein [155], as well as three MPs (TGB1, TGB2 and TGB3) [142]. TGB MP expression is mediated by two sgRNAs, sgRNAβ1 and sgRNAβ2. TGB1, the virus protein which interacts with p80-coilin, is expressed from sgRNAβ1 [152]. TGB1 is a multifunctional protein, possessing RNA helicase, ATPase and RNA binding activities. It is thought to facilitate several virus processes such as cell-to-cell movement and the establishment of a systemic infection [142]. When studied in *N. benthamiana*, WT plants infected with BSMV showed no recovery, with mild systemic symptoms persisting [140]. However, like TRV, plants with p80-coilin knockdown presented with an increase in severity of symptoms and an increase in accumulation of BSMV, implying that coilin was acting in host antivirus defence [140]. However, in contrast to TRV, coilin-dependent antivirus defence did not appear to lead to plant recovery, but instead resulted in reduced plant susceptibility to BSMV in virus-infected leaves [140].

##### Rice Stripe Virus

Rice stripe virus (RSV) is a member of the Tenuivirus genus and has a ssRNA genome consisting of four segments that encode at least seven proteins [156]. One of these proteins, p2, is responsible for long-range movement in the plant. Interestingly, p2 is targeted to both nucleoli and CBs and interacts both physically and functionally with coilin and fibrillarin without disrupting these NBs [136,137].

##### Grapevine Red Blotch-Associated Virus

Grapevine red blotch-associated virus (GrRBaV) is a single-stranded DNA virus of the Geminivirus family and contains a 3206 nucleotide genome encoding at least six proteins [135]. The V2 ORF, when expressed as a fluorescent fusion protein in *A. thaliana*, localises to the nucleoplasm and structures that may be CBs. On co-expression with a fluorescent fibrillarin construct, the V2 fusion protein becomes directed into nucleoli [135]. Thus, V2 may have the ability, like RSV p2, to target both CBs and nucleoli or may shuttle between these two NBs in complex with fibrillarin.

#### 2.5.4. Targeting Argonaute 4

De novo DNA methylation at cytosine residues is a conserved, protective epigenetic mark in eukaryotic cells, mediated in plants by RNA-directed DNA methylation (RdDM) of which Argonaut 4 (AGO4) is a component [143].

##### Tomato Yellow Leaf Curl Virus

Tomato yellow leaf curl virus (TYLCV) is a member of the Begomovirus genus of the the *Geminivirus* family. The TYLCV genome encodes six ORFs, two on the virus sense strand (V1 and V2) and four on the complementary strand (C1–4). Other ORFs have been described, e.g., C5, whose gene product is involved in virulence and suppression of gene silencing [157]. The TYCLV genome consists of circular ssDNA [158]. Mini-chromosomes are formed during infection by members of the geminivirus family by the association of virus dsDNA with host histones [159]. Minichromosomal DNA is methylated by host enzymes in order to mount an antivirus defence [160]. Hypermethylation of virus DNA led to plant recovery. Plant recovery in this context is indicative of transcriptional gene silencing (TGS) and posttranscriptional TGS (PTGS) of host pathways [161,162]. However, TYLCV protein V2 suppresses both TGS and PTGS pathways by the host, potentially via different domains, thus fulfilling an analogous role to RNA silencing suppressor proteins discovered in several other viruses [163,164]. In TYLCV-infected cells, there is an interaction between V2 and host histone deacetylase 6 (HDAC6), which resulted in interference in binding of methyltransferase 1 (MET1) to and subsequent reduction in methylation of virus DNA via TGS, thus increasing host susceptibility to TYLCV infection [165]. An additional TGS suppression capability of TYLCV V2 is present in *N. benthamania* cells, where V2 interacts with AGO4 and interrupts its binding to virus DNA.

TYLCV antivirus methylation takes place in CBs. The link between CBs and TYLCV infection was further demonstrated by the colocalisation of the V2-AGO4 complex with host CBs, which was shown to be critical for the interaction between V2 and AGO4 [143]. A similar mechanism of antivirus DNA methylation is employed by V2 of cotton leaf curl Multan virus (CLCMV), via its interaction with AGO4, although the intra-nuclear localisation of CLCMV V2 has not been reported [166].

##### Cucumber Mosaic Virus

Cucumber mosaic virus (CMV) is a member of the *Bromoviridae* family and is a tripartite ssRNA virus that encodes five proteins, of which protein 2b has host-silencing suppression functions. The 2b protein is located in Cajal bodies and nucleoli and interacts with AGO1 and 4, reducing their methylation activity and PTGS [144,145].

##### Pelargonium Line Pattern Virus

Pelargonium line pattern virus (PLPV), in addition to interacting with coilin and fibrillarin (2.5.1 Groundnut Rosette Virus), has also been reported to interfere with cytosine methylation of the promoter element of ribosomal DNA genes in infected *N. benthamiana*, leading to an overall reduction in methylation of CpG sites, resulting in an increase in pre-rRNA transcripts. This is achieved by reduction in the expression of DNA methyltransferases such as MET1 and increases in several DNA demethylases (ROS1, DML2 and DML3) [130].

## 3. Concluding Remarks: Common Threads and Future Perspectives

### 3.1. Nuclear Architecture Can Be Remodelled in Similar Ways by Diverse Viruses

Nuclear bodies (NBs) can rearrange their architecture and redistribute their core components in response to various physiological conditions. For example, in response to cellular stress or DNA damage, the nucleolus undergoes nucleolar segregation [47,48]. A similar phenomenon has been demonstrated in CBs during cellular stress [23]. Likewise, PML bodies are linked to the regulation of stress-induced sumoylation by undergoing changes in their structure [54]. A number of diverse viruses disrupt Cajal bodies in analogous ways. Following infection with ASFV, influenza A virus or ZIKV, CBs increase in number while simultaneously decreasing in volume [99,103,108]. These viruses have either RNA or DNA as their genomes, they are members of different virus families and genera and have different virus lifecycles, but appear to adopt a conserved strategy of disruption of CBs.

In some virus infections, CB rearrangement does not occur in isolation; for example, ASFV infection also causes the structural rearrangement of PML bodies as well as the production of enlarged nuclear speckles. ASFV infection is PML-dependent, since PML knockdown resulted in malformed virus factories and reduced yields of virus progeny [108], which implies that host PML bodies play an important role in virus replication. Similarly, adenoviruses reorganise CBs, PMLs and nucleoli during infection. Fragmentation of the nucleolus takes place in influenza A virus infection, providing a link between the nucleolus and CBs [104]. However, although influenza A virus NP is tightly associated with both the nucleolus and CBs in infected cells, it does not appear to induce their fragmentation [104]. Another possibility is that the rearrangement of NBs such as CBs and nucleoli occurs in response to the cellular stress that is stimulated by virus infection, as seen in host cells [122,123].

GRV in plants, and MVM and adenoviruses in mammals, all cause disruption of NBs during infection, although their mechanisms for doing so follow unique pathways. For example, MVM and GRV redistribute components of CBs to form novel bodies that also contain virus proteins. In GRV infection, CBs are reorganised, displacing coilin and fibrillarin into CB-like (CBL) structures that contain the virus ORF3 protein, which then fuse with the nucleolus. Fusion of CBLs with the nucleolus may facilitate the export of fibrillarin and ORF3 to the cytoplasm where they are used in the generation of virus RNPs. This process has been proposed to aid the establishment of a systemic infection [133,134].

In MVM, disruption of CBs arises following the formation of APAR (autonomous parvovirus-associated replication) bodies which recombine with nuclear components, including from CBs, into SMN-associated APAR bodies or SAABs; both of these bodies are sites of virus replication [100]. This infers a link between the rearrangement of CBs and the nucleolus in order to support replication of the virus.

Adenoviruses (Ad5) disrupt CBs in a novel manner, by redistributing their components into CB microfoci [97,98], potentially triggered by the dissociation of nuclear actin from CBs [96]. This rearrangement was observed to occur at a specific time in the virus life cycle and is associated with late-phase virus DNA replication [97]. CB microfoci localised to the periphery of virus replication centres, characterised by the localisation of virus protein E2A-72K [97]. A virus protein, L4-22K, forms a stable complex with p80-coilin and is located in some CB microfoci. This colocalisation strongly indicates that the virus is rearranging the host nuclear architecture in order to serve its own replication, likely in order to sequester critical factors present in CBs. PML bodies and the nucleolus are similarly disrupted upon infection by Ad5, displacing their core components [94,95]. It is also worth noting that the human Ad L4-22K protein, the TRV 16K, the PSLV MP and the RSV p2 proteins of plant viruses all form physical complexes with coilin, suggesting a common target of diverse viruses in their infectious cycles.

Taken together, this information further strengthens a proposed link between virus appropriation of genome replication machinery and rearrangement of NBs. This link may be redistributing components of these nuclear bodies to be in close proximity to virus replication sites, thus supplying them with required factors, such as snRNPs or host proteins, etc. This may be a common function of these viruses. The existence of virus proteins that are conserved between virus species, e.g., V2 in TYLCV and V2 in cotton leaf curl Multan virus that both target AGO4 [143,166] or NIa in PVA and TEV that target CBs [146], suggests that these viruses may, at least in some cases, share a common mechanism. Other steps in virus infection may be affected by the reorganisation or disruption of NBs and should be investigated, e.g., virus egress from the nucleus and generalized cytopathic effects.

### 3.2. Involvement of Nuclear Bodies in Antivirus Defence

Several viruses interact with CBs and other NBs in both plant and mammalian cells, inducing a host antivirus response. Although PML body involvement in these antimicrobial defence pathways is well documented, less is known concerning CBs. In TRV and BSMV, antivirus defence is mediated by virus interaction with coilin, resulting in recovery of the former virus and reduced severity of symptoms in the latter [140]. The interaction of TRV 16K protein with coilin was shown to facilitate both RNA silencing and SA pathways in order to coordinate an effective antivirus response [140,141]. In both TRV and BSMV, knockdown of coilin resulted in increased accumulation of the virus and an exacerbation of symptoms [140]. In addition, in plants, the TYLCV-targeted antivirus defence mechanism has been shown to encompass transcriptional and post-transcriptional gene silencing pathways, leading to plant recovery [160,161,162]. The antivirus methylation modifications required for these pathways take place in CBs [143]. TYLCV is protected from these pathways by interaction of TYLCV V2 with host HDAC6, resulting in reduced methylation of virus DNA [165].

TYLCV V2 also interacts with AGO4, a component of the RdDM pathway that localises with CBs. This interaction, and the subsequent formation of the V2-AGO4 complex, was further shown to rely on colocalisation with CBs [143]. Taken together, this implies an association between CBs and protection against several plant viruses, suggesting a functional role for CBs in antivirus defence.

In humans, HIV-1 replication is inhibited at the level of transcription by the CB component protein TOE1 [111]. During adenovirus and influenza A virus infection, NBs have been linked to contributing to an antivirus immune response. Adenoviruses target apoptosis pathways via interactions with PML body components and disrupt both PML bodies and the nucleolus, displacing their core components [94,167,168,169]. This suggests that the disruption of NBs is a core process in the virus life cycle that has evolved to protect the virus from host defence mechanisms. In influenza A virus infection, the host antivirus defence is mediated by PML bodies, which are involved in the interferon-induced antivirus pathway [107]. This is subverted by influenza A virus by accumulating NS1 and M1 in PML bodies. This proposed function for PML bodies in influenza A antivirus defence is strengthened by the discovery that host resistance to influenza A virus is facilitated by overexpression of PML [121].

It is possible that greater understanding of these mechanisms may inform the discovery of novel targets for antivirus therapies, targeting NBs such as CBs, PML bodies and nucleoli and thus disrupting the replication cycle of several viruses of plant and animal origin. It will be of considerable interest to gain greater knowledge of other virus types and the extent to which they disrupt NBs. For example, viruses that replicate only in the cytoplasm without an obvious nuclear involvement would be interesting to study, since they may induce cellular stress that might result in the reorganisation or disruption of NBs.

## Figures and Tables

**Figure 1 viruses-15-02311-f001:**
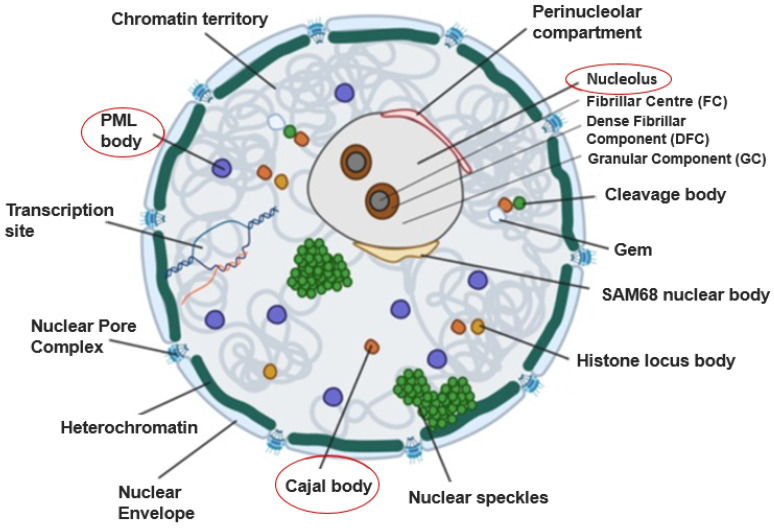
A schematic representation of the eukaryotic nucleus. Major domains and nuclear bodies (NBs) are labelled. The NBs upon which this review is focussed are nucleoli, Cajal bodies, and PML bodies (marked in red ovals). See text for details.

**Figure 2 viruses-15-02311-f002:**
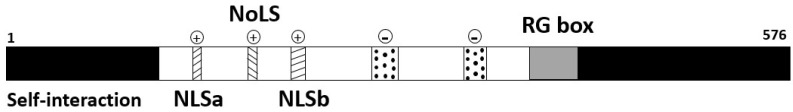
A schematic representation of the domain structure of p80-coilin. The self-interacting domain is located at the N-terminus. NLSa and NLSb refer to the two nuclear localisation signals present in coilin and NoLS refers to the nucleolar localisation signal, all of which have positive charge. The stippled boxes represent acidic serine-rich patches. The RG box domain is located at the C-terminus. The C-terminus of coilin is folded into a Tudor domain. Modified from [63].

**Table 1 viruses-15-02311-t001:** Viruses that infect members of the animal kingdom and interact with Cajal bodies as well as other relevant nuclear bodies.

Species Infected by Virus	Name of Virus	Virus Protein/Nucleic Acid Sequence Element That Interacts with Cajal Bodies	Cajal Body Protein That Interacts with Virus	Function of Interaction	Other Nuclear Bodies That Interact with Virus (Along with CBs)	References
*Aves* (Birds), principally *Gallus gallus domesticus* (Chicken)	Marek’s Disease Virus (MDV) (Avian alphaherpesvirus)	Meq/Meq/vIL8	p80-Coilin	Unknown	Nucleoplasm, nucleolus	[92,93]
Several mammalian species, including *Homo sapiens* (Human)	Adenovirus Type 5 (Ad5)	L4-22K	p80-coilin	Ad5 infection disrupts CBs, redistributed into CB microfoci; L4-22K forms complex with p80-coilin. Depletion of p80-coilin reduces export of virus mRNAs from nucleus in infected cells	PML bodies (E4-ORF3 11 kDa), PML bodies (E1A)	[94,95,96,97,98]
Several animal species, including *Homo sapiens* (Human)	Zika Virus (ZIKV)	NS5	p80-coilin	Unknown, disrupt CBs, resulting in an increased number of smaller CBs	Unknown	[99]
*Murinae* (Murine), principally *Mus* *musculus* (Mouse)	Minute Virus of Mice (MVM)	NS1	SMN	Disrupt components of CBs, recombined into SAABs (sites of virus replication)	Unknown	[100]
*Homo sapiens* (Human)	Human Cytomegalovirus (HCMV)	UL3 and UL30	Unknown	Unknown, expression of UL3 and UL30 decreased number of CBs in cell	PML bodies	[101]
Several animal species, including *Homo sapiens* (Human)	Influenza A virus	NP	Unknown	Unknown, expression of NP results in an increased number of smaller CBs	PML bodies, nucleolus (virus NS1)	[102,103,104,105,106,107]
*Sus scrofa domesticus* (Swine)	African Swine Fever Virus (AFSV)	Unknown	Unknown	Unknown, disrupt CBs, results in an increased number of smaller CBs	Nuclear speckles, PML bodies	[108,109,110]
*Homo sapiens* (Human)	Human Immunodeficiency Virus (HIV)	TAR (Transactivation Response RNA Element)	TOE1	Host antivirus defence	Unknown	[111,112,113]

**Table 2 viruses-15-02311-t002:** Viruses that infect members of the plant kingdom and interact with Cajal bodies as well as other relevant nuclear bodies.

Species Infected by Virus	Name of Virus	Virus Protein That Interacts with Cajal Bodies	Cajal Body Protein That Interacts with Virus	Function of Interaction	Other Nuclear Bodies That Interact with Virus (Along with CBs)	References
*Nicotiana benthamiana*	Citrus tristeza virus (CTV)	P23	Fibrillarin	Suppression of RNA silencing, enhancing systemic infection and virus accumulation	Nucleolus	[129]
*Nicotiana benthamiana*	Pelargonium line pattern virus (PLPV)	P37	Fibrillarin, coilin	Suppression of RNA silencing, overcoming host defence, modulation of methylation of ribosomal DNA promoter	Nucleolus	[130,131]
*Solanum tuberosum* (Potato)	Potato Virus A (PVA)	NIa	Fibrillarin	Unknown, reduced fibrillarin resulted in reduced accumulation of PVA	Nucleolus	[132]
*Arachis hypogaea* (Groundnut)	Groundnut Rosette Virus (GRV)	ORF3	Fibrillarin	Long-distance movement, establishing systemic infection, stabilisation of virus RNA	Nucleolus	[133,134]
*Vitis* (Grapevine)	Grapevine Red Blotch-associated Virus (GrRBaV)	V2	Fibrillarin, coilin	Unknown	Nucleolus, inclusions in the nucleoplasm	[135]
*Oryza sativa* (Rice)	Rice Stripe Virus (RSV)	P2	Coilin	Unknown	Nucleolus (via fibrillarin to establish systemic infection)	[136,137]
*Poaceae* (Grass)	Poa Semilatent Virus (PSLV)	TGBp1	Coilin	Cell-to-cell movement, long distance transport, establishing a systemic infection	Nucleolus	[138,139]
A wide variety of species	Tobacco Rattle Virus (TRV)	16 K	Coilin	Host antivirus defence (RNA and SA silencing pathways)	Nucleolus	[140,141]
*Hordeum vulgare* (Barley)	Barley Stripe Mosaic Virus (BSMV)	TGB1	Fibrillarin, coilin	Cell-to-cell movement of virus during infection	Unknown	[142]
*Solanum ycopersicum* (Tomato)	Tomato Yellow Leaf Curl Virus (TYLCV)	V2	AGO4	Suppression of host antivirus defence (inhibit TGS and PTGS pathways)	Nucleoplasm	[143]
A wide variety of species	Cucumber Mosaic Virus (CMV)	2b	AGO4	RNA-directed DNA methylation	Nucleolus	[144,145]

## Data Availability

No new data was created in this work.

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
