# Peer review of "Viruses and Cajal Bodies: A Critical Cellular Target in Virus Infection?"

_viruses, 2023, doi:10.3390/v15122311_

Round 1
Reviewer 1 Report
Comments and Suggestions for Authors
The paper entitled “Viruses and Cajal bodies: a critical target in virus infection?” reviewed the current knowledge of the importance in virus infection of the nuclear bodies with a focus on the Cajal bodies. This exhaustive review showed that the nuclear bodies which contained biomolecular condensates including proteins and different types of RNA are targeted by both animal and plant virus proteins. As a result of these virus interactions, the pathogenicity is either increased or decreased.
Comments:
1. There are some minor modifications regarding the form of the review which might help reading this review:
1.1 Name of species are in bold with the size of the character bigger than the main text: page3; 6; 7…
1.2 Lane 4 page 5: protein (for? I guess) the mouse
1.3 Lane 32 page 5: mutation in what ???
1.4 Why named a chapter 2.3.1 if there is no chapter 2.3.2?
2. Table 1: You mention TAR as a virus protein that interacts with Cajal bodies! As written in the main text, TAR is a RNA motif which interacts with the Tat viral transactivator. Are you sure that the reference (111) mentioned in this table 1 is appropriate for HIV-1?
You claimed that no other nuclear bodies interact with HIV-1. However it has been shown that viral DNA and viral RNA clusters are associated with speckles factors (Rensen E et al, 2020). Moreover it is also proposed that some unspliced HIV-1 RNA might be stored in nuclear niches named paraspeckles (Zhang Q et al, 2013)
Comments on the Quality of English Languageenglish language is correct
Author Response
Reviewer # 1
The paper entitled “Viruses and Cajal bodies: a critical target in virus infection?” reviewed the current knowledge of the importance in virus infection of the nuclear bodies with a focus on the Cajal bodies. This exhaustive review showed that the nuclear bodies which contained biomolecular condensates including proteins and different types of RNA are targeted by both animal and plant virus proteins. As a result of these virus interactions, the pathogenicity is either increased or decreased.
Comments:
- There are some minor modifications regarding the form of the review which might help reading this review:
Thank you for the thorough review of the paper and your constructive suggestions.
1.1 Name of species are in bold with the size of the character bigger than the main text: page3; 6; 7…
This has been changed throughout the paper.
1.2 Lane 4 page 5: protein (for? I guess) the mouse
Yes this is correct and …the mouse…has been added for clarity.
1.3 Lane 32 page 5: mutation in what ???
This has been altered to …..point mutations or deletions in the SMN1 gene
1.4 Why named a chapter 2.3.1 if there is no chapter 2.3.2?
Good point! This seemed to follow the Viruses style but we will ask the Editorial Team for advice in this - in the meantime we have left as is.
- Table 1: You mention TAR as a virus protein that interacts with Cajal bodies! As written in the main text, TAR is a RNA motif which interacts with the Tat viral transactivator.
Thank you for pointing out this ambiguity. We have changed the entry in Table 1 to define TAR as an RNA element and also altered the title of the column. We have also discussed emphasised this point in the section on HIV.
Are you sure that the reference (111) mentioned in this table 1 is appropriate for HIV-1?
Well spotted, thank you! This has been changed to reference 100 in Table 1.
You claimed that no other nuclear bodies interact with HIV-1. Howeve,r it has been shown that viral DNA and viral RNA clusters are associated with speckles factors (Rensen E et al, 2020). Moreover, it is also proposed that some unspliced HIV-1 RNA might be stored in nuclear niches named paraspeckles (Zhang Q et al, 2013)
Thank you for directing our attention to this point. We have addressed the reviewer’s remarks at the end of the section on HIV and cited the papers by Rensen and Zhang.
Reviewer 2 Report
Comments and Suggestions for Authors
The manuscript provides a concise overview of the significance of nuclear bodies (NBs) in eukaryotic cells and their dynamic role in cellular processes and virus-host interactions. The authors effectively emphasize the importance of Cajal bodies (CBs) in eukaryotic cells with a summary of the extensive literature during virus-host interactions, encompassing both plant and animal systems.
The breadth of the content is commendable, offering a well-structured and logically coherent narrative. However, there are some inaccuracies in the content that require attention. The absence of proper referencing is a notable concern, and the instances of incorrect references should be very carefully addressed.
Major issues:
l Incorrect citation of references in the main text.
a) Section 2.5.4 Targeting Argonaute 4, the first paragraph.
REF #151 is incorrect for the AGO4 and RdDM. Please add a correct reference here.
b) Section 3.2. Involvement of Nuclear Bodies in Anti-virus Defence, the second paragraph.
Ref #159 should be Ref #151.
1. Incorrect citation numbering
There are so many mistakes in Table 2. Please carefully double-check all the Refs cited in the Table 1 and Table2, as well as main text. Some mistakes in the table 2 are listed below:
a) For P23 from CTV, reference should be #130
b) For NIa from PVA, reference should be #128
c) For ORF3 from GRV, reference should be #134-135.
d) For V2 from GrRBaV, reference should be #150. And the reference does not show the interaction between V2 and Fibrillarin, since there is only proof for colocalization of V2 and Fib.
e) For P2 from GrRBaV, reference should be #148-149. And P2 interacts with both Coilin and Fibrillarin.
f) For 16K from TRV, reference should be #142-143.
g) For TGB1 from BSMV, reference should be #146 (Hijacking of the nucleolar protein fibrillarin by TGB1 is required for cell-to-cell movement of Barley stripe mosaic virus). And viral protein is “TGB1”, not unknown. The CB proteins should be Fibrillarin, not Coilin. The function of interaction should be “cell-to-cell movement of virus during infection”.
h) For V2 from TYLCV, reference should be #151.
i) For 2b from CMV, reference should be #161-162.
2. Inaccuracies or imprecise content:
a) 2.5.4.1 Tomato Yellow Leaf Curl Virus, sentence “The TYLCV genome encodes six ORFs, two on the virus sense strand (V1 and V2) and four on the complementary strand (C1-4).”
First, the writing of Tomato Yellow Leaf Curl Virus does not require capitalizing the initial letter. Secondly, TYLCV has traditionally been considered to have six classical viral proteins, however, new viral proteins have been continuously discovered. For example, “The novel C5 protein from tomato yellow leaf curl virus is a virulence factor and suppressor of gene silencing.
b) 3.2. Involvement of Nuclear Bodies in Anti-virus Defence, sentences “In addition, in plants, the TYLCV-targeted anti-virus defence mechanism has been shown …shown to occur in CBs [151].”
Please be more precise in referring to the geminivirus family and the specific species TYLCV. For instance, references 154-156 correspond to the Geminivirus family, while reference 151 corresponds to the TYLCV one species.
c) Table 2: GRBaV should be GrRBaV,
Minor comments:
Inconsistent spacing and font formatting, which may hinder the overall presentation.
1. "Remove the extra spaces scattered throughout the text."
Section 1.1: Cajal bodies, third line of the first paragraph, and the fourth paragraph.
Section 1.3: PML bodies, second paragraph.
2. Inconsistencies in font formatting and size. e.g.:
Section 1.1: The final paragraph, "In Arabidopsis thaliana, components of the plant gene..." and "In A. thaliana and tobacco BY-2 cells...";
Section 1.4.1: "Germline knockout of p80-coilin in mice [4], zebrafish [3], and A. thaliana revealed...";
Section 1.4.6: Second and third paragraphs.
l Inconsistent use of abbreviations for “histone deacetylase 6”
Section 2.5.4.1 Tomato Yellow Leaf Curl Virus:histone deacetylase 6 (HDAC6);
Section 3.2. Involvement of Nuclear Bodies in Anti-virus Defence: histone deacetylase 6.
Comments on the Quality of English Language
Overall, the manuscript is well-written, the authors' proficient use of the English language is evident throughout the manuscript.
Author Response
Reviewer # 2
This review focuses on the observed interactions between either animal and plant viral proteins with nuclear bodies (NBs) and, more particularly, with Cajal bodies (CBs). The biological significance of those interactions, if known, is described and discussed. The review is in general well written and summarizes the state of the art of an issue with many unkowns. I have some concerns that should be addressed before publication:
Thank you for your detailed review of the paper and for identifying important viruses that were overlooked in our original version.
-Page 1. “For example, fibrillarin and Nopp140 are present in both NBs..” . It is not clear to which NBs the sentence is referring to.
This has been corrected in the text which now says…present in CBs and nucleoli.
-Page 3-4 and beyond. Different fonts (especially for species names) in some sentences. Unify.
This has been corrected throughout the text. Thank you.
-Page 6. Small non-coding RNAs rea known as sncRNAs instead of snRNAs; please, correct and avoid confusions with small nuclear RNAs (snRNAs).
Thank you for raising this – this has been corrected.
-Table 2 and main text. Include data on protein p37 of Pelargonium line pattern virus which interacts with fibrillarin and coilin (Pérez-Cañamas et al., 2022, Plants (Basel), 11(15):1903). This protein also interacts with AGO4 (and AGO1) though the impact of this interaction on TGS is unclear (Pérez-Cañamas et al., 20220 Biology (Basel) 9(5):91)
We have added data on PLPV to Table 2 and discussed this virus along with the other Tombusvirus, GRV, in terms of the interaction of p37 of PLPV with coilin and fibrillarin, and in the AGO4 section in terms of the modulation of rDNA promoter methylation. We have added the two papers on PLPV to the reference list.
- A recent review on CBs in plants should be included (Taliansky et al., 2023; Plant Cell.
35(9):3214-3235)
This reference has been added to the paragraph on plant CBs (part of the longer section on CBs)
Reviewer 3 Report
Comments and Suggestions for Authors
This review focuses on the observed interactions between either animal and plant viral proteins with nuclear bodies (NBs) and, more particularly, with Cajal bodies (CBs). The biological significance of those interactions, if known, is described and discussed. The review is in general well written and summarizes the state of the art of an issue with many unkowns. I have some concerns that should be addressed before publication:
-Page 1. “For example, fibrillarin and Nopp140 are present in both NBs..” . It is not clear to which NBs the sentence is referring to.
-Page 3-4 and beyond. Different fonts (especially for species names) in some sentences. Unify.
-Page 6. Small non-coding RNAs rea known as sncRNAs instead of snRNAs; please, correct and avoid confusions with small nuclear RNAs (snRNAs).
-Table 2 and main text. Include data on protein p37 of Pelargonium line pattern virus which interacts with fibrillarin and coilin (Pérez-Cañamas et al., 2022, Plants (Basel), 11(15):1903). This protein also interacts with AGO4 (and AGO1) though the impact of this interaction on TGS is unclear (Pérez-Cañamas et al., 20220 Biology (Basel) 9(5):91)
- A recent review on CBs in plants should be included (Taliansky et al., 2023; Plant Cell. 35(9):3214-3235)
Author Response
Reviewer # 3
The review by Lettin, Erbay and Blair is a sweeping survey of viruses that interact with nuclear bodies, especially Cajal bodies. The scope of this work is rather breathtaking and covers dozens of viruses of plants and animals. The text is very well written, with a clear and consistent style. This work summarized a vast number of observations and places them in an accessible (albeit long) form. Consequently, this manuscript would be of value for those seeking connections among viruses or connections among pathogens that target nuclear processes.
I find little about the document to criticise but offer three comments for the authors’ consideration.
Thank you for your detailed review of the paper and for raising interesting general issues regarding our interpretation of data in the literature.
First, is the “perinuclear compartment” correctly labelled in Figure 1? Why is that not a “peri-nucleolar compartment?”
Thank you, you are correct. We have altered the diagram (Fig 1) accordingly.
Second, the disruption of nuclear bodies is portrayed as though this is a “purposeful” action of the virus, typically suggested to defeat an anti-viral response. Could many of these effects simply be part of the overall cytopathic effect and be little more than a side-effect of the virus recruiting and using cellular machinery? I would be inclined to welcome comments that recognize or dispute this possible scenario.
Thank you for raising this interesting point. We have tended in the review to ascribe purpose to, for example, the loss, disaggregation or modulation of CB in certain virus-infected cells. However we have to concede that at present other interpretations are possible, including processes such as cytopathic effect, virus egress from the nucleus etc. We have added a sentence at the end of section 3.1 that addresses these possibilities.
Third, the synopsis at the end of the document suggested that viruses that replicate only in the cytoplasm would be of interest to study as they may have less of a “need” to reorganize the various nuclear bodies. This is a reasonable notion but it seems that surely some observations have been made on this subject? Perhaps these are not the focus of the report, but I find it hard to imagine that so many cytoplasmic viruses have been studied without some observation on the cytopathic effect in the nucleus. Moreover, if the cellular anti-viral response is transcriptional, would not all viruses that suppress the this anti-viral response affect events in the nucleus, irrespective of whether the virus lifecycle involved the nucleus or not?
In preparing this review we conducted an extensive literature search for papers using broad search terms such as “viruses and cajal bodies” etc. We did not find any papers on cytoplasmic viruses and NB re-arrangement or disruption. Following this comment, we have done more targeted searching combining the above terms with “picornavirus”, “flavivirus” etc, namely viruses known to replicate in the cytoplasm without an obvious nuclear involvement. However such searches did not reveal any papers. At present, we conclude that such studies have either not been performed or reported. The second point is also a good one but we have not found any papers that bear on this point.
Reviewer 4 Report
Comments and Suggestions for Authors
The review by Lettin, Erbay and Blair is a sweeping survey of viruses that interact with nuclear bodies, especially Cajal bodies. The scope of this work is rather breathtaking and covers dozens of viruses of plants an animals. The text is very well written, with a clear and consistent style. This work summarized a vast number of observations and places them in an accessible (albeit long) form. Consequently, this manuscript would be of value for those seeking connections among viruses or connections among pathogens that target nuclear processes.
I find little about the document to criticise but offer three comments for the authors’ consideration.
First, is the “perinuclear compartment” correctly labelled in Figure 1? Why is that not a “peri-nucleolar compartment?”
Second, the disruption of nuclear bodies is portrayed as though this is a “purposeful” action of the virus, typically suggested to defeat an anti-viral response. Could many of these effects simply be part of the overall cytopathic effect and be little more than a side-effect of the virus recruiting and using cellular machinery? I would be inclined to welcome comments that recognize or dispute this possible scenario.
Third, the synopsis at the end of the document suggested that viruses that replicate only in the cytoplasm would be of interest to study as they may have less of a “need” to reorganize the various nuclear bodies. This is a reasonable notion but it seems that surely some observations have been made on this subject? Perhaps these are not the focus of the report, but I find it hard to imagine that so many cytoplasmic viruses have been studied without some observation on the cytopathic effect in the nucleus. Moreover, if the cellular anti-viral response is transcriptional, would not all viruses that suppress the this anti-viral response affect events in the nucleus, irrespective of whether the virus life cycle involved the nucleus or not?
Author Response
Reviewer # 4
The manuscript provides a concise overview of the significance of nuclear bodies (NBs) in eukaryotic cells and their dynamic role in cellular processes and virus-host interactions. The authors effectively emphasize the importance of Cajal bodies (CBs) in eukaryotic cells with a summary of the extensive literature during virus-host interactions, encompassing both plant and animal systems.
The breadth of the content is commendable, offering a well-structured and logically coherent narrative. Overall, the manuscript is well-written, the authors' proficient use of the English language is evident throughout the manuscript.
However, there are some inaccuracies in the content that require attention. The absence of proper referencing is a notable concern, and the instances of incorrect references should be very carefully addressed.
In conclusion, while the manuscript effectively covers the significant aspects of CBs and virus-host interactions, attention to issues such as formatting, content accuracy, and proper referencing is necessary to enhance the overall quality and credibility of the work
Thank you for your positive comments about the paper and your detailed analysis of the text and Tables. We apologise for the errors in the citation references and appreciate the opportunity to correct them.
Major issues:
⚫ Incorrect citation of references in the main text.
- a) Section 2.5.4 Targeting Argonaute 4, the first paragraph.
REF #151 is incorrect for the AGO4 and RdDM. Please add a correct reference here.
- b) Section 3.2. Involvement of Nuclear Bodies in Anti-virus Defence, the second paragraph.
Ref #159 should be Ref #151.
We have addressed and corrected all of these points regarding referencing although, as some other reviewers requested addition of papers, the numbering in the reference list may have changed.
⚫ Incorrect citation numbering
There are so many mistakes in Table 2. Please carefully double-check all the Refs cited in the Table 1 and Table2, as well as main text. Some mistakes in the table 2 are listed below:
- a) For P23 from CTV, reference should be #130
- b) For NIa from PVA, reference should be #128
- c) For ORF3 from GRV, reference should be #134-135.
- d) For V2 from GrRBaV, reference should be #150. And the reference does not show the interaction between V2 and Fibrillarin, since there is only proof for colocalization of V2 and Fib.
All of these references have been corrected although numbering may have changed due to insertion of new references as requested by reviewers. Regarding point 5 above, we have qualified our statement to take account of the reviewer’s comment.
- e) For P2 from GrRBaV, reference should be #148-149. And P2 interacts with both Coilin and Fibrillarin.
- f) For 16K from TRV, reference should be #142-143.
- g) For TGB1 from BSMV, reference should be #146 (Hijacking of the nucleolar protein fibrillarin by TGB1 is required for cell-to-cell movement of Barley stripe mosaic virus). And viral protein is “TGB1”, not unknown. The CB proteins should be Fibrillarin, not Coilin. The function of interaction should be “cell-to-cell movement of virus during infection”.
- h) For V2 from TYLCV, reference should be #151.
- i) For 2b from CMV, reference should be #161-162.
All of these points above have been corrected.
⚫ Inaccuracies or imprecise content:
- a) 2.5.4.1 Tomato Yellow Leaf Curl Virus, sentence “The TYLCV genome encodes six ORFs, two on the virus sense strand (V1 and V2) and four on the complementary strand (C1-4).”
First, the writing of Tomato Yellow Leaf Curl Virus does not require capitalizing the initial letter. Secondly, TYLCV has traditionally been considered to have six classical viral proteins, however, new viral proteins have been continuously discovered. For example, “The novel C5 protein from tomato yellow leaf curl virus is a virulence factor and suppressor of gene silencing.
A sentence has been added to the text and an appropriate reference added. The Table 2 entry for this virus has been corrected.
- b) 3.2. Involvement of Nuclear Bodies in Anti-virus Defence, sentences “In addition, in plants, the TYLCV-targeted anti-virus defence mechanism has been shown …shown to occur in CBs [151].”
This has been re-phrased and corrected.
Please be more precise in referring to the geminivirus family and the specific species TYLCV. For instance, references 154-156 correspond to the Geminivirus family, while reference 151 corresponds to the TYLCV one species.
We have changed the discussion of references 151 to 156 to reflect this comment.
- c) Table 2: GRBaV should be GrRBaV,
This has been changed to the correct abbreviation.
Minor comments:
Inconsistent spacing and font formatting, which may hinder the overall presentation.
These errors have been altered – please see above
⚫ "Remove the extra spaces scattered throughout the text."
Section 1.1: Cajal bodies, third line of the first paragraph, and the fourth paragraph.
Section 1.3: PML bodies, second paragraph.
These have been corrected.
⚫ Inconsistencies in font formatting and size.
Section 1.1: The final paragraph, "In Arabidopsis thaliana, components of the plant gene..." and "In A. thaliana and tobacco BY-2 cells...";
Section 1.4.1: "Germline knockout of p80-coilin in mice [4], zebrafish [3], and A. thaliana revealed...";
Section 1.4.6: Second and third paragraphs.
⚫ Inconsistent use of abbreviations for “histone deacetylase 6”
Section 2.5.4.1 Tomato Yellow Leaf Curl Virus histone deacetylase 6 (HDAC6);
Section 3.2. Involvement of Nuclear Bodies in Anti-virus Defence: histone deacetylase 6.
All of these points have been corrected.